# Exploratory Study on the Foliar Incorporation and Stability of Isotopically Labeled Amino Acids Applied to Turfgrass

**Rachel M. McCoy [1,2]** , **George W. Meyer [1,2]** , **David Rhodes [1]** , **George C. Murray [3]** , **Thomas G. Sors [4] and Joshua R. Widhalm [1,2,*]**

1   Department of Horticulture and Landscape Architecture, Purdue University, West Lafayette, IN 47907, USA; mccoy26@purdue.edu (R.M.M.); meyer335@purdue.edu (G.W.M.); davidrhodes124@gmail.com (D.R.)
2   Center for Plant Biology, Purdue University, West Lafayette, IN 47907, USA
3   EnP Investments, LLC, 2001 West Main Street, Mendota, IL 61342, USA; george@foliarpak.com
4   Purdue Institute of Inflammation, Immunology and Infectious Disease, Purdue University, West Lafayette, IN 47907, USA; tsors@purdue.edu
*   Correspondence: jwidhalm@purdue.edu; Tel.: +1-765-496-3891

**Abstract:** There is increasing interest in the use of amino acid-based biostimulant products due to their reported abilities to improve a number of quality characteristics in a variety of specialty crops. However, when it comes to the foliar application of amino acids to turfgrass, there are still many basic questions about their uptake forms and incorporation into cellular metabolism. In this study, we shed light on the fate of amino acids exogenously applied to turfgrass foliage through a series of time-course, isotopic-labeling studies in creeping bentgrass (*Agrostis stolonifera* L.) leaves. Using both $^{15}N$-labeled and $^{15}N,^{13}C$ double-labeled L-glutamate applied exogenously to creeping bentgrass foliage, we measured the uptake of glutamate and its integration into $\gamma$-aminobutyric acid (GABA) and L-proline, two amino acids with known roles in plant stress adaptation. Our results demonstrate that glutamate is rapidly absorbed into creeping bentgrass foliage and that it is utilized to produce GABA and proline. Based on the labeling patterns observed in the endogenous pools of glutamate/glutamine, GABA, and the proline from applied glutamate-$[^{13}C_5{}^{15}N_1]$, we can further conclude that glutamate is predominantly taken up intact and that mineralization into other forms of nitrogen is a minor fate. Taken together, the collective findings of this study provide evidence that amino acids exogenously applied to turfgrass foliage can be rapidly absorbed, and serve as stable sources of precursor molecules to be integrated into the metabolism of the plant.

**Keywords:** biostimulants; amino acids; isotopic labeling; turfgrass

## 1. Introduction

The use of biostimulants to promote quality traits in specialty crops has gone up over the last decade [1]. With an estimated annual growth rate of more than 10% each year, the projected market for biostimulants is estimated to be at $4.9 billion by 2025 [2]. Biostimulants is a broad term referring to extracts, lysates, purified natural compounds or microorganisms that are applied to crops in small amounts to enhance aspects like health, resiliency, and/or vigor [1,3] but whose primary role is not to fertilize or protect against pathogens [4].

Amino acids and small peptide-based biostimulants have received increased attention for their positive effects on plant performance [5]. Whilst externally applied amino acids are poorly taken up by roots because of competition with soil microbes, foliar application has the potential to improve availability due to reduced competition [6]. As a result, amino acids are emerging in many foliarly

applied products marketed to golf course superintendents and sports turf managers with claims of enhanced growth, greening, and increased resistance to stress. Despite the substantial sales of such products from a variety of companies in the turfgrass market, there have been limited studies on the uptake by and the fate of amino acids in turfgrass foliage. Using [15]N-labeled glycine, L-glutamate, and L-proline, it was previously demonstrated that the nitrogen from these applied amino acids was absorbed into creeping bentgrass (*Agrostis stolonifera* L.) foliage to similar degrees as other nitrogen fertilizer forms [7]. Assuming that no mineralization into other transportable forms of nitrogen occurred on the leaf's surface, this suggests that amino acids can be directly taken up through bentgrass foliage. This now raises questions about the metabolic fate of exogenously applied amino acids once inside the plant. The objective of this study was to determine the uptake form, stability, and incorporation of amino acids exogenously applied into metabolism in turfgrass foliage. To accomplish this, we conducted a series of exploratory tracer studies using [15]N- and [15]N,[13]C-labeled glutamate, applied exogenously to bentgrass foliage, and we measured their integration into endogenous amino acid pools and derived metabolites.

## 2. Materials and Methods

### 2.1. Plant Growth Conditions, General Experimental Procedures, and Reagents

Turfgrass used in this experiment was PennTrio Bentgrass (Tee-2-Green Corporation, Hubbard, OR, USA) which is a creeping bentgrass (*Agrostis stolonifera* L.) mix that contains equal parts Penncross, Penneagle, and Pennlinks. Turfgrass was grown in a controlled environment in 8″ pots at 23–24 °C with an average humidity of 45% under daylight spectrum fluorescent lighting with 12-h days. Plants were watered weekly and fertilized once at germination with a fertilizer containing 12% nitrogen, 6% phosphorus ($P_2O_5$), 6% potassium ($K_2O$), and micronutrients boron, copper, iron, manganese, and zinc. Unlabeled amino acid standards were purchased from Sigma-Aldrich (St. Louis, MO, USA). Stable isotopes were purchased from Cambridge Isotopes (Tewksbury, MA, USA). All other reagents were purchased from Fisher Scientific (Pittsburgh, PA, USA). For gas chromatography-mass spectrometry (GC-MS) experiments, an Agilent 7890B GC (Agilent Technologies, Santa Clara, CA, USA) connected with an Agilent 5966A mass spectrometer (Agilent Technologies, Santa Clara, CA, USA) were used. All analyses were done using Agilent Chemstation software.

### 2.2. Stable Isotope Labeling of Turfgrass

Labeling was conducted by spraying the foliage of the potted plants with a mixture of each stable-isotopically labeled amino acid in water at a rate of 804 L per hectare, at the concentrations indicated below. The first trial used 10 mM glutamate-[[15]$N_1$] (Cambridge Isotopes) with sampling at 0, 1, 4, 8, 24, and 48 h post application. In the second trial, 4 mM glutamate-[[13]$C_5$-[15]$N_1$] was applied and sampling occurred at 0, 0.25, 0.5, 1, 4, 8, 24, 48, and 72 h post application. At each timepoint, the aboveground tissue was cut and rinsed to remove any residue and then the leaves were transferred directly into methanol to quench metabolism, and stored at 4 °C until extraction.

### 2.3. Extraction and Quantification of Amino Acids

Amino acids were extracted according to a protocol adapted from Rhodes et al. [8] using a ration of 10 mL methanol for every 500 mg of creeping bentgrass leaves. Extracts were spiked with 25 μL of 10 mM α-aminobutyrate, vortexed well, and then incubated in the dark at 4 °C for 2 d to extract metabolites. Next, for every 10 mL methanol, 5 mL chloroform and 6 mL water were added and incubated for 1 h at room temperature to allow phase separation. The aqueous phase was collected and evaporated to dryness under $N_2$ gas using a Techne sample concentrator. The dried aqueous phase was resuspended in 1 mL of water and applied to a Dowex-50-$H^+$ 200 mesh column. The column was washed with 7 mL water, and amino acids were eluted from the column with 6 mL of 6 M $NH_4OH$ and dried. Amino acids were derivatized for GC-MS analysis, as described previously [9], with 1 μL

of each derivatized sample being analyzed by GC-MS on an Agilent 19091s-433 HP-5MS capillary column (30 m × 0.25 mm; film thickness 0.25 μm) as described previously [8]. Labeling percentage was calculated by dividing the intensity of the shifted molecular ion by the sum of the shifted and unshifted ion and corrected for natural isotope abundance. See Supplementary Tables S1 and S2 for masses analyzed for each labeled and unlabeled amino acid.

## 3. Results and Discussion

### 3.1. Nitrogen from Foliar Applied Glutamate is Incorporated into Proline and γ-aminobutyric acid (GABA)

To investigate whether amino acids are absorbed by turfgrass leaves and incorporated into cellular metabolism, we measured time course labeling in the endogenous pools of glutamate and some major glutamate-derived amino acids from glutamate-$[^{15}N_1]$ applied to the foliage of creeping bentgrass. In addition to serving as the precursor for the synthesis of chlorophylls and proteins, glutamate functions as a hub metabolite in plant amino acid metabolism (Figure 1). Glutamate is a substrate for producing L-glutamine from ammonia; it serves as the primary α-amino donor for aminotransferases involved in synthesizing multiple amino acids, and its carbon skeleton and amino group are directly incorporated into L-arginine, L-proline, and γ-aminobutyric acid (GABA) [10]. The accumulation of GABA, a non-proteinogenic amino acid found ubiquitously in plants, functions in adaptive responses to mitigate plant stress, including defense against drought and insect herbivory [11]. The overproduction of proline was also demonstrated to be a metabolic response involved in plant stress tolerance. Proline functions as an osmolyte to maintain cell turgor, stabilizes membranes to prevent electrolyte leakage, and helps prevent oxidative bursts by lowering the concentrations of reactive oxygen species [12]. Therefore, because enhanced resiliency to environmental stresses underlies one of the major purported benefits of amino acid-based biostimulant products, we focused on labeling in GABA and proline from glutamate-$[^{15}N_1]$ exogenously applied to creeping bentgrass foliage.

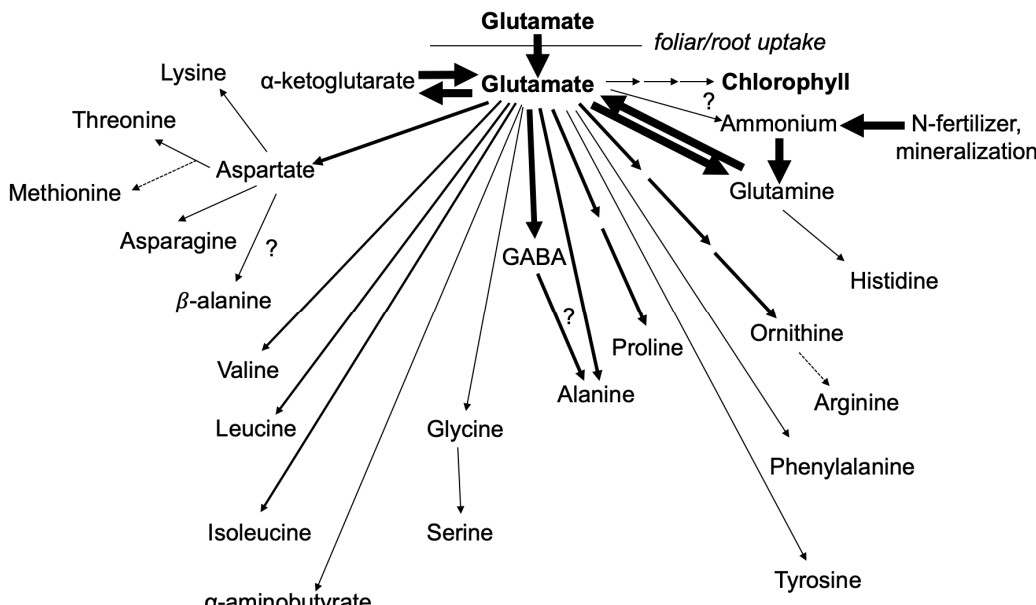

**Figure 1.** Glutamate occupies a central position in plant amino acid metabolism. The arrows indicate the multiple fates for the carbon backbone and/or amino group of glutamate in plant cells. The arrow thickness approximately correlates with relative flux toward each metabolite. The arrows labeled with a question mark (aspartate to β-alanine (aspartate decarboxylase), GABA to alanine (GABA: pyruvate aminotransferase), and glutamate to ammonium (glutamate dehydrogenase)) denote metabolic fates that are controversial.

To ensure that the endogenous precursor pool was labeled highly enough to detect possible labeling in GABA and proline, we first examined labeling in glutamate by looking at the glutamate/glutamine pool. Note that in the current sample preparation protocol, glutamine is converted into glutamate during derivatization, so the two amino acids are quantified together by GC-MS as glutamate. Within 1 h of foliar application with glutamate-[$^{15}N_1$], the glutamate/glutamine pool was labeled by 60% and remained constant over the 48-h experiment (Figure 2). The pool of GABA, which is formed via the irreversible decarboxylation of glutamate in plant cytoplasm by glutamate decarboxylase (GDC) [10], was labeled by 29% within 1 h of glutamate-[$^{15}N_1$] application, increased to over 40% labeled 4 h post application, and then remained relatively constantly labeled for the duration of the experiment (Figure 2). The rapid incorporation of glutamate into GABA is consistent with the observation that the expression of the gene encoding GDC in rice roots increased nearly 10-fold in response to exogenous application of glutamate [13]. Labeling in proline, whose biosynthesis from glutamate can take place in chloroplasts or cytoplasm [14], was in comparison expectedly delayed (Figure 2). The proline pool was labeled by 12% 4 h after application with glutamate-[$^{15}N_1$], increased to 23% labeled by 8 h, and then remained constant until 48 h. Taken together with the fact that glutamate must be present in cytoplasm to produce GABA and in the cytoplasm or chloroplast to synthesize proline, these data are consistent with not only glutamate-[$^{15}N_1$] being absorbed into the foliage of creeping bentgrass, but also with it being taken up by cells where it can be utilized to produce metabolites with well-established roles in plant stress adaptation.

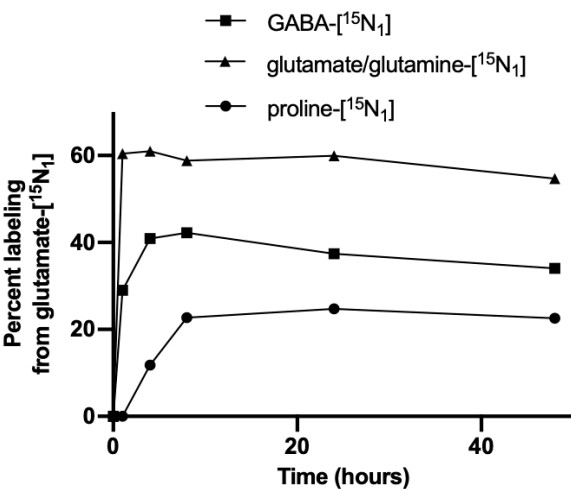

**Figure 2.** Time course of percent labeling of glutamate/glutamine-[$^{15}N_1$], GABA-[$^{15}N_1$], and proline-[$^{15}N_1$], from 10 mM glutamate-[$^{15}N_1$] applied to the foliage of creeping bentgrass (*Agrostis stolonifera* L.).

### 3.2. The Carbon Skeleton from Foliar Applied Glutamate is also Incorporated into Proline and GABA

In a previous labeling study by Stiegler et al. [7], it was found that the uptake of nitrogen from glycine, glutamate, and proline into creeping bentgrass foliage is equal to or less than that of nitrogen from urea. Thus, it is possible that glutamate-[$^{15}N_1$] applied to creeping bentgrass foliage in the current study was mineralized on the leaf surface and that ammonia-[$^{15}N$] was absorbed and then re-assimilated into glutamine/glutamate (Figure 1) before being used to synthesize GABA-[$^{15}N_1$] and proline-[$^{15}N_1$] (Figure 2). To definitively determine whether glutamate-[$^{15}N_1$] was taken up intact or mineralized before absorption, we performed the same time course labeling experiment with glutamate-[$^{13}C_5$$^{15}N_1$]. By using double-labeled glutamate, in which the nitrogen and all carbon atoms are labeled, it is possible to differentiate between the uptake of mineralized ammonia-[$^{15}N$] and the intact amino acid.

Similarly to what was observed with glutamate-[$^{15}N_1$] (Figure 2), applied glutamate-[$^{13}C_5$$^{15}N_1$] rapidly labeled the glutamate/glutamine pool (Figure 3A). The predominant form detected was

the fully intact form, glutamate/glutamine-[$^{13}C_5{}^{15}N_1$], which represented approximately 55% of the total pool and remained relatively constant for the duration of the experiment. The second most abundantly labeled form detected was glutamate/glutamine-[$^{13}C_5$]. It was found to represent approximately 10% of the total pool and then attenuated to nearly 0% by 24 h after application. This form would a priori derive from the metabolism of glutamate-[$^{13}C_5{}^{15}N_1$] to α-ketoglutarate-[$^{13}C_5$] that is transaminated back to glutamate-[$^{13}C_5$] with an unlabeled nitrogen. The least abundant form detected was glutamate/glutamine-[$^{15}N_1$], representing less than 3% of the total pool by 1 h post application and rapidly decreasing thereafter. This form likely results from the labeled nitrogen of absorbed glutamate-[$^{13}C_5{}^{15}N_1$] being used to transaminate an unlabeled α-ketoglutarate to produce glutamate-[$^{15}N_1$]. This form could also originate if applied glutamate-[$^{13}C_5{}^{15}N_1$] was mineralized on the leaf surface to produce ammonia-[$^{15}N$] that was absorbed and then re-assimilated back into amino acid metabolism to produce glutamate-[$^{15}N_1$] (Figure 1). Regardless of how it was formed, because glutamate-[$^{15}N_1$] accounted for such a small fraction of the total glutamate/glutamine pool compared to the $^{13}C$-labeled forms, this suggests that the intact amino acid was the predominant form absorbed by turfgrass foliage.

Next, we examined whether the glutamate-[$^{13}C_5{}^{15}N_1$] applied to creeping bentgrass foliage labeled GABA and proline like what was observed with glutamate-[$^{15}N_1$] (Figure 2). Peak labeling in GABA occurred 1 h after application with glutamate-[$^{13}C_5{}^{15}N_1$], though labeling was already detectable at 15 min (Figure 3B). Unlike the first experiment (Figure 2), there was a decrease in labeled GABA pools following the initial peak (Figure 3B). This likely reflects the fact that less glutamate-[$^{13}C_5{}^{15}N_1$] was administered. Previous work in rice by Kan et al. [13] showed that the expression of the gene encoding GDC displays a sensitive dosage-dependent induction in response to glutamate. The subsequent decline and increase in GABA labeling is likely related to the incorporation of GABA into the GABA shunt, a bypass pathway in which the GABA produced in the cytoplasm is imported into the mitochondria, where it is converted to succinate that can enter the tricarboxylic acid (TCA) cycle. The GABA shunt is the major source of succinate in foliage during the day (reviewed in Michaeli and Fromm, 2015 [15]).

The most abundantly labeled form of GABA detected was GABA-[$^{13}C_4{}^{15}N_1$], which represented approximately 9% of the total pool. This isotopic form likely originated from decarboxylation of glutamate-[$^{13}C_5{}^{15}N_1$], the predominant labeled form found in the glutamate pool (Figure 3A). The other isotopic forms of GABA detected after 1 h, GABA-[$^{13}C_4$] and GABA-[$^{15}N_1$], represented 7.5% and 3.3% of the total pool, respectively (Figure 3B). Because GABA-[$^{15}N_1$] is a priori synthesized from glutamate-[$^{15}N_1$], the observation that GABA-[$^{15}N_1$] was the least abundant labeled form present is consistent with glutamate-[$^{15}N_1$] being the minor form in the glutamate pool (Figure 3A). Along the same lines, proline-[$^{13}C_5{}^{15}N_1$] and proline-[$^{13}C_5$] were more abundant than proline-[$^{15}N_1$]; however, like in the previous experiment (Figure 2), labeling was delayed, peaking at 24 h post application with glutamate-[$^{13}C_5{}^{15}N_1$] (Figure 3C). Thus, in all cases, the double-labeled $^{13}C$ and $^{15}N$ isotopes and the single-labeled $^{13}C$ isotopes of glutamate, GABA, and proline were more abundant than the single-labeled $^{15}N$ isotopic forms (Figure 3A–C). These data imply that intact amino acids are taken up by turfgrass foliage rather than being mineralized to other transportable forms of nitrogen. The data also indicate that once inside the plant, exogenously applied amino acids are imported into cells where they can be rapidly and directly incorporated into metabolism.

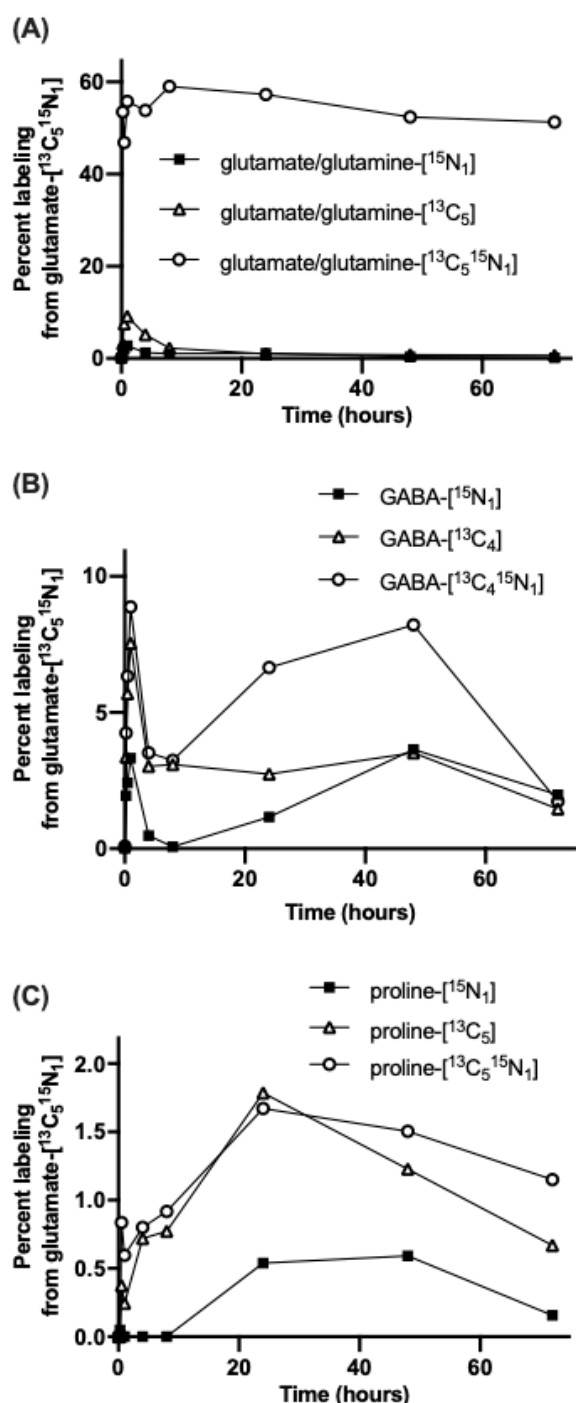

**Figure 3.** Time course of percent labeling of single- and double-labeled isotopic forms of glutamate/glutamine (**A**), GABA (**B**) and proline (**C**) from 4 mM glutamate-$[^{13}C_5{}^{15}N_1]$ applied to foliage of creeping bentgrass (*Agrostis stolonifera* L.).

## 4. Conclusions

In this exploratory study, we investigated questions about the uptake forms and the incorporation of exogenously applied amino acids on turfgrass foliage. Through time course labeling studies with glutamate-$[^{15}N_1]$ and glutamate-$[^{13}C_5{}^{15}N_1]$, we demonstrated that glutamate is rapidly absorbed intact into creeping bentgrass leaves and directly utilized as a precursor to synthesize GABA and proline, two well-studied glutamate-derived metabolites with roles in plant stress adaptation. Our results also provide evidence that the mineralization of glutamate into other nitrogen forms is likely a minor fate of

the amino acids applied to the foliage, though future work measuring the formation and foliar uptake of other nitrogen forms should be performed to independently investigate this question. Furthermore, the labeling in the endogenous pools of glutamate/glutamine remained stable for 72 h, the latest point measured in this study. Taken together, the collective findings of our work suggest that amino acids applied to turfgrass foliage, like those in some specialty turf care products, can be rapidly absorbed and serve as stable sources of precursor molecules to be integrated into the metabolism of the plant.

**Supplementary Materials:** The following are available online at http://www.mdpi.com/2073-4395/10/3/358/s1, Table S1: Fragments of each labeled and unlabeled amino acid from glutamate-[$^{15}N_1$] as analyzed by GC-MS; Table S2: Fragments of each labeled and unlabeled amino acid from glutamate-[$^{13}C_5^{15}N_1$] as analyzed by GC-MS.

**Author Contributions:** Conceptualization, D.R., T.G.S, G.C.M., and J.R.W.; methodology, D.R., T.G.S, G.C.M., and J.R.W.; formal analysis, R.M.M., G.W.M., D.R., and J.R.W.; investigation, R.M.M., G.W.M., and D.R.; resources, G.C.M and J.R.W.; writing—original draft preparation, R.M.M., G.W.M., D.R., G.C.M., and J.R.W.; writing—review and editing, R.M.M., G.W.M., D.R., G.C.M., T.G.S., and J.R.W.; visualization, R.M.M., G.W.M., D.R., and J.R.W.; supervision, D.R., G.C.M., and J.R.W.; project administration, D.R., G.C.M., T.G.S., and J.R.W.; funding acquisition, D.R., G.C.M., and J.R.W. All authors have read and agreed to the published version of the manuscript.

**Funding:** This research was funded in part by EnP Investments, LLC (Mendota, IL USA), start-up funds from Purdue University to J.R.W., and by the USDA National Institute of Food and Agriculture Hatch Project number 177845.

**Acknowledgments:** We thank Kyle Ladenburger of EnP Investments, LLC for his assistance in performing amino acid applications and collecting tissue.

**Conflicts of Interest:** G.C.M. is the president of EnP Investments, LLC (manufacturer of the Foliar-Pak brand, Mendota, IL USA) where he is also the chief formulator and inventor. He was involved in the design of the study and collection of the samples, and decision to publish, but was not involved in the analyses or interpretation of data. The remaining authors declare no competing financial or non-financial interests.

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
