# Peer review of "Exploratory Study on the Foliar Incorporation and Stability of Isotopically Labeled Amino Acids Applied to Turfgrass"

_agronomy, doi:10.3390/agronomy10030358_

Round 1

Reviewer 1 Report

Review manuscript agronomy 732581 - Exploratory Study on the Foliar Incorporation and Stability of Isotopically Labeled Amino Acids Applied to Turfgrass - by McCoy et al.

The brief report describes the uptake and incorporation of exogenously applied isotopic-labelled amino acid glutamate into the amino acid pool of turfgrass. Based on the increased use of biostimulants in agriculture research of the effectiveness and mode of action of biostimulants are important. This brief research paper is a very good contribution. The introduction provides all necessary background information and the intention of the study. Material and Methods contain all necessary information about the experimental setup. Results and discussion are mostly conclusive and well written. The authors argue that the GABA [15N1] is a priori synthesized from glutamate[15N1], on what is the seen increase up to 3 % after a drastic reduction after 6-8h based on. All labelled GABA forms decrease drastically after a short time peak with a slow increase afterwards beside of GABA[13C4]. It would be interesting for the reader to learn about the metabolic background of this behaviour. The same for the labelled proline forms.

Author Response

Reviewer Point 1: "The authors argue that the GABA [15N1] is a priori synthesized from glutamate[15N1], on what is the seen increase up to 3 % after a drastic reduction after 6-8h based on. All labelled GABA forms decrease drastically after a short time peak with a slow increase afterwards beside of GABA[13C4]. It would be interesting for the reader to learn about the metabolic background of this behaviour. The same for the labelled proline forms."

Response to Point 1: The reviewer points out an interesting observation for which multiple explanations are possible. Based on what we see in Figure 2 and in Figure 3B, rapid labeling in the GABA pool was detected within the first 30 minutes after feeding labeled glutamate. This seems to be most consistent with the observation made by others that expression of the gene encoding glutamate decarboxylase (GDC1, the enzyme that converts glutamate to GABA) is upregulated almost 10-fold in rice roots fed with glutamate (Kan et al, 2017; BMC Genomics 18:186). In the same study, expression of the GDC1 gene was found to have a sensitive dosage-dependent induction in response to glutamate with expression peaking 30 min post exposure and then attenuating to near pretreatment levels by 24 hr. The labeling data presented in our study reflect a similar trend: labeling in the GABA pool rapidly increases after feeding labeled glutamate and then decreases. What is interesting in Figure 3B, however, is that after the initial spike and decline in GABA labeling, the percent labeling increases again based on measurements at 24 and 48 hr. (Note, the decline at 72 hours likely has to do with exhausting the labeled precursor pool). This may suggest an effect of the GABA shunt: a bypass pathway in which GABA produced in the cytoplasm is imported into the mitochondria where it is converted to succinate that can enter the TCA cycle. The GABA shunt is the major source of succinate in foliage during the day (reviewed in Michaeli and Fromm, Front. Plant Sci. 6:419). Thus, the combined effect of an exogenous glutamate source with the diurnal fluctuation in the activity of the GABA shunt could offer hypotheses to explore in the future. Labeling in the proline pool, on the other hand, was comparatively delayed. This likely reflects the fact that while glutamate is converted to GABA in one step, the conversion to proline involves multiples steps, and hence more coordinated regulation.

While we agree with the reviewer that it would be interesting for the reader to learn about the metabolic background of this behavior, we want to be careful about over speculating. Therefore, we have added some context to the discussion without trying to over extend the interpretation of our data. Please see:

New lines 139-144: "The pool of GABA, which is formed via irreversible decarboxylation of glutamate in plant cytoplasm by glutamate decarboxylase (GDC) [10], was labeled by 29% within 1 h of glutamate-[15N1] application, increased to over 40% labeled 4 h post application, and then remained relatively constantly labeled for the duration of the experiment (Figure 2). The rapid incorporation of glutamate into GABA is consistent with the observation that expression of the gene encoding GDC in rice roots increases nearly 10-fold in response to exogenous application of glutamate [13]."

New lines 196-203: "Unlike the first experiment (Figure 2), there was a decrease in labeled GABA pools following the initial peak (Figure 3B). This likely reflects the fact that less glutamate-[13C515N1] was administered. Previous work in rice by Kan et al. [13] showed that expression of the gene encoding GDC displays a sensitive dosage-dependent induction in response to glutamate. The subsequent decline and increase in GABA labeling is likely related to incorporation of GABA into the GABA shunt, a bypass pathway in which GABA produced in the cytoplasm is imported into the mitochondria where it is converted to succinate that can enter the TCA cycle. The GABA shunt is the major source of succinate in foliage during the day (reviewed in Michaeli and Fromm, 2015 [15])."

Reviewer 2 Report

The objective  of study should me more clear. Author should rewrite it.

Units shoudl be changed in the International Units System: for example line 74 page 2.

Author Response

Reviewer Point 1: "The objective  of study should me more clear. Author should rewrite it."

Response to Point 1: Thank you for pointing out the need to clarify the study objective. We have now changed the Introduction from:

"This now raises questions about the metabolic fate of exogenously applied amino acids once inside the plant, including determining their stability and incorporation into cellular metabolism. To address these questions and to investigate the uptake form of amino acids into turfgrass foliage, we have conducted a series of exploratory tracer studies using 15N- and 15N,13C-labeled glutamate applied exogenously to bentgrass foliage and measured their integration into endogenous amino acid pools and derived metabolites."

To (see changes on lines 52-58, page 2):

"This now raises questions about the metabolic fate of exogenously applied amino acids once inside the plant. The objective of this study was to determine the uptake form, stability, and incorporation of exogenously applied amino acids into metabolism in turfgrass foliage. To accomplish this, we conducted a series of exploratory tracer studies using 15N- and 15N,13C-labeled glutamate applied exogenously to bentgrass foliage and measured their integration into endogenous amino acid pools and derived metabolites."

Reviewer Point 2: "Units shoudl be changed in the International Units System: for example line 74 page 2."

Response to Point 2: Thank you for catching this. We have changed 86 gallons per acre to 804 L per hectare. Line 74, page 2.